# PHYSICS-INFORMED RADIAL BASIS FUNCTION NETWORKS AND KOLMOGOROV-ARNOLD NETWORKS

**D.A. Stenkin**
Penza State University
Russia, Penza, Krasnaya Street, 40
stynukin@mail.ru

**V.I. Gorbachenko**
Penza State University
Russia, Penza, Krasnaya Street, 40
gorvi@mail.ru

## ABSTRACT

Physics-informed neural networks are trained by minimizing the loss function, which is the sum of the squares of the residuals of the equation or system of equations being solved. Such networks do not require grid construction, which is especially important when solving inverse boundary value problems and problems with a complex solution domain. We use radial basis function networks with a Gaussian function. Physics-informed radial basis function networks are easier to train than fully connected networks. They allow one to analytically obtain formulas for the gradient of the loss function. A special feature of our approach to training networks based on radial basis functions is the adjustment of not only the weights, but also the parameters of the radial basis functions, which does not require the selection of parameters of the radial basis functions and accelerates the training process. Algorithms for solving direct and inverse boundary value problems, an algorithm for solving a system of differential equations for modeling the Kovasznay flow have been developed. Programs have been developed that use various algorithms for training physics-informed radial basis function networks.

## 1 INTRODUCTION

Neural networks are applicable for solving boundary value problems described by partial differential equations (1; 2). This possibility is based, first of all, on the fact that neural networks are universal function approximators. Cybenko's theorem states that an artificial neural network with one hidden layer can approximate any continuous function of many variables with any accuracy (3). Hornik (4) proved that a multilayer perceptron with an arbitrary number of hidden layers and an arbitrary number of neurons in each layer can approximate any continuous function with any accuracy on a compact set. It was also proved that such approximation is possible for arbitrary activation functions if they are bounded, continuous, non-polynomial and non-constant. A variational approach is also used to solve partial differential equations on neural networks: the solution to the problem is found by minimizing the error functional of the neural network. The residuals of the approximate solution obtained by the network at a certain set of trial points inside, on the boundary of the solution domain and, possibly, at points with additional conditions are used as the error functional. Estimating the norm of this residual gives a better interpretable estimate of the accuracy of the problem solution than estimates of the accuracy of the solution of grid methods, for which only the order of accuracy of the approximation and the remainder of the solution of the system of grid equations are known. Trial points are usually located randomly in the solution domain. Currently, a specific class of neural networks has been formed for solving partial differential equations on neural networks - physics-informed neural networks. Physics-informed neural networks (5; 6; 7) include in the network structure a mathematical model of some physical phenomenon, for example, partial differential equations. When solving direct problems, PINN do not require examples with known values for training. Such networks solve the problem of low data availability in many scientific and engineering problems where traditional machine learning methods are ineffective. Additional conditions can also be used, for example, when solving inverse problems, experimental information about the solution at individual points of the region is used. In physics-informed neural networks,

the mathematical model is a regulating factor that improves the quality of the solution. Radial basis function networks are successfully used to solve partial differential equations. Radial basis function networks (8; 2; 9) are simpler than fully connected networks, since they contain only two layers - a layer of radial basis functions (10) and a linear layer. The simplicity of the network architecture simplifies the training of radial basis function networks, for example, it allows analytical calculation of the gradients of the error functional. The authors have developed second-order gradient learning algorithms for radial basis function networks, in which not only the network weights are configured, but also the parameters of the radial basis functions. The efficiency of the developed algorithms has been demonstrated by solving direct (11; 12) and inverse (13; 14) boundary value problems. Physics-informed radial basis function networks (15; 14; 16; 17) are an effective tool for solving partial differential equations. Radial basis function networks have some similarities with Kolmogorov-Arnold networks. Kolmogorov-Arnold networks are based on the Kolmogorov-Arnold theorem (18), which states that any continuous multivariate function $f(x_1, x_2, ..., x_d)$ defined on a bounded domain can be represented as a finite composition of continuous one-dimensional functions and addition operations. Mathematically, this is expressed as:

$$f(x_1, x_2, ..., x_d) = \sum_{q=1}^{2d+1} \Phi_q \left( \sum_{p=1}^{d} \psi_{q,p}(x_p) \right),$$

where: $\Phi_q$ and $\psi_{q,p}$ are continuous one-dimensional functions. In (19), R. Hecht-Nielsen proposed a neural network that directly reflects the Kolmogorov-Arnold representation, where the hidden layer computes the internal sums $\sum_{p=1}^{d} \psi_{q,p}(x_p)$ and the output layer computes the external sums $\sum_{q=1}^{2d+1} \Phi_q(.)$. But R. Hecht-Nielsen did not propose methods for finding the specific functions $\Phi_q$ and $\psi_{q,p}$, which makes his network difficult to implement directly. Therefore, such networks have not been developed.

Kolmogorov-Arnold networks (20) are also based on the Kolmogorov-Arnold theorem and introduce significant innovations in architecture and training. Unlike the Hecht-Nielsen approach, Kolmogorov-Arnold networks replace fixed activation functions with trainable cubic basis splines (B-splines) in which the basis function weights are adjustable. Other basis functions can also be used.

Radial basis function networks are similar to Kolmogorov-Arnold networks in that in both types of networks, the inputs of the first layer are nonlinearly transformed using basis functions and added together. In physics-informed radial basis function networks implementing the regression problem, the second nonlinear layer is absent. In the one-dimensional case, as shown in (21), a complete analogy between Kolmogorov-Arnold networks and radial basis function networks can be traced. In the multidimensional case, there is no analogy, since the basis functions are functions of many variables, which violates the conditions of the Kolmogorov-Arnold theorem. There are no data comparing radial basis function networks and Kolmogorov-Arnold networks. Solving partial differential equations on physics-informed neural networks is a neural network approximation of the problem solution. It is easier to approximate a multidimensional result function with multidimensional functions than with one-dimensional ones, so it can be assumed that with an increase in the problem dimension and in the case of inhomogeneous media, radial basis function networks will outperform Kolmogorov-Arnold networks. To enhance the positive qualities of radial basis function networks, it is necessary to adapt radial basis functions to the specific features of the problem being solved, which requires adjusting not only the network weights, but also the parameters of the radial basis functions. Such adjustment is implemented in the radial basis function network training algorithms we propose.

The aim of this work is to develop algorithms for solving various boundary value problems on networks of radial basis functions trained by gradient algorithms with the adjustment of not only the network weights, but also the parameters of the radial basis functions.

## 2 DEVELOPMENT OF ALGORITHMS FOR LEARNING PHYSICS-INFORMED RADIAL BASIS FUNCTION NETWORKS

The output of the radial basis function network is described by the expression

$$v(x, y) = \sum_{k=1}^{n_{RBF}} \omega_k \varphi_k(x, y),$$

where $n_{RBF}$ — number of radial basis functions (or number of neurons), $\omega_k$ — weight of the $k$ th neuron, $\varphi_k(x, y)$ — value of the $k$ th radial basis function at point $x$.

To solve the problems in this article, Gauss functions were used as radial basis functions, which have the form

$$\varphi_k(x, y) = e^{-\frac{(x - c_{k1})^2 + (y - c_{k2})^2}{2a_k^2}}$$

where $c_{k1}$ and $c_{k2}$ — centers of functions, $a_k$ — width of functions.

Consider a model direct problem in operator form having the form

$$Lu(\mathbf{x}) = f(\mathbf{x}), \ x \in \Omega,$$

$$Bu(\mathbf{x}) = p(\mathbf{x}), \ x \in \partial\Omega,$$

where $u$ — desired solution; $L$ — differential operator; operator $B$ sets the boundary conditions; $\Omega$ — solution area; $\partial\Omega$ — regional boundary; $f$ and $p$ — known functions.

Let us consider the effectiveness of the Levenberg-Marquardt diagonal method for solving the Poisson equation

$$\frac{\partial^2 u}{\partial x^2} + \frac{\partial^2 u}{\partial y^2} = f(x, y), \ (x, y) \in \Omega,$$

$$u(x, y) = p(x, y), \ (x, y) \in \partial\Omega,$$

where $\Omega$ — estimated area; $\partial\Omega$ — the boundary of the computational domain; $f = \sin(\pi x) \cdot \sin(\pi y), p = 0$.

The error function can be written as

$$\frac{1}{2} \sum_{i=1}^{N} \left( \frac{\partial^2 u_i}{\partial x^2} + \frac{\partial^2 u_i}{\partial y^2} - f_i \right)^2 + \frac{\lambda}{2} \sum_{j=1}^{K} (u_j - p_j)^2$$

where $N$ – number of internal test points, $K$ – number of boundary test points, $\lambda$ – penalty multiplier.

The network was trained to solve the Poisson equation using various methods: gradient descent, Nesterov, Levenberg-Marquardt, and diagonal approximation of the Hessian. The gradient method converges at a linear rate. And Newton's method has a quadratic convergence rate. But to implement Newton's method, it is necessary to find the Hessian, the inverse Hessian matrix, and solve the system. These are quite labor-intensive tasks. Especially finding the inverse matrix. Therefore, some modification can be considered. We will consider the Hessian to be a diagonal matrix. We equate all mixed partial derivatives to zero. Thanks to this, we can easily find the inverse matrix. We can also adjust the convergence rate of the algorithm. This will help to circumvent the problem of linear minimization of the error functional when the second derivatives become close to zero.

Due to the convenient structure of neural networks, the formulas of the second derivatives for the Poisson equation were obtained analytically. The values of the elements located on the main diagonal of the Hessian for this problem can be obtained based on the following formula

$$\frac{\partial^2 I}{\partial \omega_k^2} = \sum_{i=1}^{N} \left( e^{-\frac{(x_i - c_{k1})^2 + (y_i - c_{k2})^2}{2a_k^2}} \left( \frac{(x_i - c_{k1})^2 + (y_i - c_{k2})^2 - 2a_k^2}{a_k^4} \right) \right)^2 +$$

$$\lambda \sum_{j=1}^{K} \left( e^{-\frac{(x_i - c_{k1})^2 + (y_i - c_{k2})^2}{2a_k^2}} \right)^2$$

where $c_{k1}$ and $c_{k2}$ — centers of functions, $a_k$ — width of functions.

The vector of weight coefficients in the Hessian diagonal approximation method is calculated using the formula

$$\omega_i^{k+1} = \omega_i^k - \eta(\frac{\partial^2 I^k}{\partial \omega_i^{k2}} + \mu)^{-1}\frac{\partial I^k}{\partial \omega_i^k}$$

where $\frac{\partial I^k}{\partial \omega_i^k}$ — gradient vector components, $\eta$ and $\mu$ — selected coefficients.

The Hessian diagonal approximation method achieved an accuracy of $10^{-3}$ in an average of 250 iterations. The gradient descent method did not achieve an accuracy of $10^{-3}$. The Nesterov method achieved an accuracy of $10^{-3}$ in an average of 500 iterations.

Let us consider the efficiency of the modified Levenberg-Marquardt method with the adjustment of the parameters of radial basis functions for solving new model problems describing processes in a piecewise homogeneous medium.

The model problem describing processes in a piecewise homogeneous medium is presented in the form of the following equation

$$\frac{\partial}{\partial x}(\sigma_i(x,y)\frac{\partial u}{\partial x}) + \frac{\partial}{\partial y}(\sigma_i(x,y)\frac{\partial u}{\partial y}) = f(x,y),$$
$$(x,y) \in \Omega, \, i = 1,2,$$
$$u(x,y) = p(x,y), \, (x,y) \in \partial\Omega,$$

where $\Omega$ — estimated area; $\partial\Omega$ — the boundary of the computational domain; $f = \sin(2\pi x) \cdot \sin(\pi y), p = 0, \sigma_i$ — a function describing the properties of the environment. The solution area is a square with unit side length, divided at $x = 0,5$ by a vertical line into two subareas with different values of $\sigma_i, i - 1, 2$. $x \in [0;1], y \in [0;1]$.

The problem is decomposed into two problems for regions 1 and 2 taking into account the conjugation conditions.

For area 1:
$$\frac{\partial}{\partial x}(\sigma_1(x,y)\frac{\partial u_1}{\partial x}) + \frac{\partial}{\partial y}(\sigma_1(x,y)\frac{\partial u_1}{\partial y}) = f_1(x,y),$$
$$(x,y) \in \Omega, \, i = 1,2,$$
$$u_1(x,y) = p_1(x,y), \, (x,y) \in \partial\Omega_1,$$
where $\Omega_1$ — design area 1; $\partial\Omega_1$ — design area boundary 1.

For area 2:
$$\frac{\partial}{\partial x}(\sigma_2(x,y)\frac{\partial u_2}{\partial x}) + \frac{\partial}{\partial y}(\sigma_2(x,y)\frac{\partial u_2}{\partial y}) = f_2(x,y),$$
$$(x,y) \in \Omega, \, i = 1,2,$$
$$u_2(x,y) = p_2(x,y), \, (x,y) \in \partial\Omega_2,$$
where $\Omega_2$ — design domain 2; $\partial\Omega_2$ — design area boundary 2.

At the interface between the media, the interface conditions must be fulfilled.

$$u_1\mid_S = u_2\mid_S, \sigma_1\frac{\partial u_1}{\partial x}\mid_S = \sigma_2\frac{\partial u_2}{\partial x}\mid_S$$

where $S$ - regional boundary.

The radial basis functions were Gaussian functions. As a result of training, not only the vector of weights, but also the parameters of radial basis functions were tuned, since it is important to tune all parameters of the radial basis function network. The Levenberg-Marquardt algorithm is used to train the radial basis function network (22). The Levenberg-Marquardt method is equivalent to the fast confidence region method (23), but does not require solving the conditional optimization problem

at each training iteration. The single vector of weights and parameters of the radial basis function network at iteration $k$ is found by the formula $\theta^{k+1} = \theta^k + \triangle\theta^{k+1}$, in which the vector of correction parameters $\theta^k$ is found from the solution of the system of linear algebraic equations

$$(\mathbf{J}_{k-1}^T \mathbf{J}_{k-1} + \mu_k \mathbf{E})\triangle\theta^k = -\mathbf{g}_{k-1},$$

where $\mathbf{J}_{k-1}$ and $\mathbf{J}_k$ — Jacobi matrices calculated in $k-1$ and $k$ iterations, $\mathbf{E}$ — unit matrix, $\mu_k$ — regularization parameter changing at each training step, $g = \mathbf{J}^T r$ — is the gradient vector of the loss function along the parameter vector $\theta$, $r$ —vector of unconstraints in internal and boundary trial points.

Two radial basis function networks are used for the solution, since each of the networks describes its own area. During network initialization, the centers of radial basis functions were located on a uniform grid. The vectors of weights and widths were initialized with random values close to zero. Currently, there are no theoretical estimates of the optimal number of neurons and test points. The authors' experimental studies have shown that there is an optimal number of neurons and test points for each task. With a small number of neurons and test points, low decision accuracy is obtained or the learning process does not converge. With a large number of neurons and test points, high accuracy of the solution is ensured, but the network operation time to obtain a solution increases significantly. The number of neurons, test points and initial values of the parameters of the radial basis functions were selected experimentally. As a result of the experiments, the following values were obtained. The number of neurons used to solve a direct boundary value problem for each of the two regions is 64. The initial values of the components of the width vector are 0.2. The number of internal test points for two regions is N=80. The number of boundary test points for two regions is K=80. The number of test points on the median line is 20. The coordinates of the collocation points were generated as random numbers evenly distributed in the solution area.

Iterations were performed until the average square error was reached, equal to $10^{-10}$. The radial basis function network was trained by the Levenberg-Marquardt method. The solution is achieved in an average of 570 iterations. Due to the random initialization of the network, the number of iterations varies in different experiments.

The inverse boundary value problem is described by a partial differential equation

$$\frac{\partial}{\partial x}(k(\mathbf{x})\frac{\partial u}{\partial x}) + \frac{\partial}{\partial y}(k(\mathbf{x})\frac{\partial u}{\partial y}) = f(\mathbf{x}), \ \mathbf{x} \in \Omega,$$

where $k(\mathbf{x})$ — continuous unknown function describing the environment, $\mathbf{x}$ — the coordinate vector, $\Omega$- solution area. Without knowing the position of the interface boundary, it is impossible to model a piecewise homogeneous function of the medium. Therefore, when solving the inverse problem, the piecewise homogeneous function describing the medium was approximately replaced by a continuous function. The boundary conditions have the form

$$Bu(\mathbf{x}) = p(\mathbf{x}), \ x \in \partial\Omega,$$

where $\partial\Omega$ — the border of the region. It is necessary to find the solution $u$ and the $k(\mathbf{x})$ function by the solution known in some set of points

$$u(\mathbf{z}) = \psi(\mathbf{z}), \ \mathbf{z} \in Z, \ Z \subset \Omega$$

moreover, $\psi(\mathbf{z})$ in real problems is the result of measurement with some error. The solution is made on radial basis function networks. To obtain the values of additional conditions, a direct problem has been solved previously on a radial basis function network. The direct boundary value problem was a differential equation for a piecewise homogeneous medium with coefficients equal to 2 and 5.5. We use parametric optimization (24). To do this, we approximate the unknown $k(\mathbf{x})$ function with a radial basis function network

$$k_{RBF}(\mathbf{x}) = \sum_{m=1}^{M_k} \omega_m^k \varphi_m^k(\mathbf{x}; \mathbf{p}_m^k),$$

where $M_k$ the number of radial basis functions (or the number of neurons), $\mathbf{p}_m^u$ — weights and vectors of parameters of radial basis functions $\varphi_m^u$.

The solution of the $u$ direct problem, in which the k($\mathbf{x}$) function is approximated by the network, is found by the second network of radial basis functions

$$u_{RBF}(\mathbf{x}) = \sum_{m=1}^{M_u} \omega_m^u \varphi_m^u(\mathbf{x}; \mathbf{p}_m^u),$$

where $M_u$ the number of radial basis functions (or the number of neurons), $\mathbf{p}_m^u$ — weights and vectors of parameters of radial basis functions $\varphi_m^u$.

Two networks were used to solve the inverse problem. The network $k_{RBF}$ for approximating the desired function of the medium. And the network $u_{RBF}$ to approximate the solution. The error functional was the sum of the squares of the residuals of the approximate solution at the test points inside, at the boundary of the solution domain and at the points of additional conditions.

For an incorrect inverse problem, regularization was required. The method of iterative regularization was the Morozov condition (25). The network has been trained so far

$$\sum_{m=1}^{S} [u(\mathbf{x}_m) - \psi(\mathbf{x}_m)]^2 > S\delta^2,$$

where $\delta$ — the absolute error of measuring the solution at the points of additional conditions.

Gaussian functions were used as radial basis functions. The number of neurons, test points and hyperparameters of learning were selected experimentally. For both problems, the number of radial basis functions is 64. For both networks, 80 test points were used inside the solution area. For the $u_{RBF}$ network, there are 80 test points on the boundary and 40 additional condition points. The initial values of the components of the width vector for the two regions are 0.3. All penalty coefficients of the error functional are 100. To check the efficiency of the algorithm, a problem was considered that has an absolute error of measuring the solution at the points of additional conditions equal to 0.0001. To solve this problem, the Morozov condition was used. The error functional value of $10^{-5}$ is achieved in an average of 290 iterations.

In Fig. 1 the desired function of the medium is presented, obtained as a result of solving the inverse boundary value problem, which models processes in a piecewise homogeneous medium.

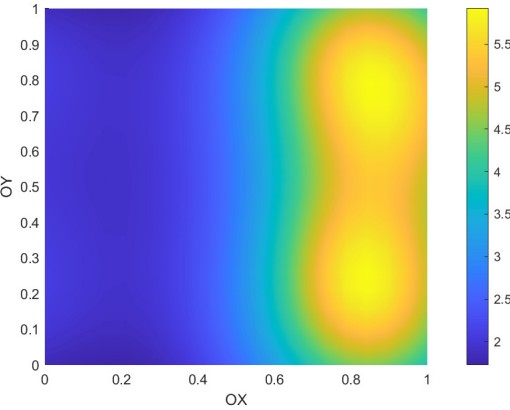

Figure 1: The environment function obtained as a result of solving the inverse boundary value problem

Judging by the graph, the neural network has successfully restored an unknown function of the medium in the process of solving the inverse boundary value problem.

The solution of the problems is implemented in the MATLAB system, which provides ease of programming and experimentation with the program and powerful graphical means of presenting the results of the solution. A feature of the implementation is the analytical calculation of the gradients of the loss function according to the parameters of the network and the Jacobi matrix, which is not difficult to implement due to the simplicity of the network structure.

# 3   MODELING A HYDRODYNAMIC PROBLEM ON PHYSICS-INFORMED RADIAL BASIS FUNCTION NETWORKS

We will demonstrate the effectiveness of physics-informed radial basis function networks for solving a model problem of hydrodynamics. For clarity, we can consider the Kovasznay current (26), since it has an analytical solution.

$$u(x, y) = -e^{(-\lambda x)} \cos(2\pi y),$$

$$v(x, y) = -\frac{\lambda}{2\pi} e^{(-\lambda x)} \sin(2\pi y),$$

$$p(x, y) = -\frac{1}{2} e^{(-2\lambda x)},$$

where parameter $\lambda$ is defined by the following expression

$$\lambda = \sqrt{\frac{Re^2}{4} + 4\pi^2} - \frac{Re}{2},$$

where $Re$ – Reynolds number.

The Kovasznay flow is a two-dimensional stationary flow. This problem is described by the Navier-Stokes system of equations. Two–dimensional stationary Navier-Stokes equations for an incompressible medium have the form:

$$\frac{\partial u}{\partial x} + \frac{\partial v}{\partial y} = 0, \ (x, y) \in \Omega,$$

$$u\frac{\partial u}{\partial x} + v\frac{\partial u}{\partial y} = -\frac{\partial p}{\partial x} + \frac{1}{Re}\left(\frac{\partial^2 u}{\partial x^2} + \frac{\partial^2 u}{\partial y^2}\right), \ (x, y) \in \Omega,$$

$$u\frac{\partial v}{\partial x} + v\frac{\partial v}{\partial y} = -\frac{\partial p}{\partial y} + \frac{1}{Re}\left(\frac{\partial^2 v}{\partial x^2} + \frac{\partial^2 v}{\partial y^2}\right), \ (x, y) \in \Omega,$$

where $\Omega$ — is the calculated area, $u$ – first velocity component, $v$ – second velocity component, $p$ – pressure. Dirichlet conditions are set at the boundary of the region.

The Kovasznay flow is the movement of the flow through the lattice. Kovasznay flow does not have an initial condition. If the flow is stationary, then the streamlines coincide with the trajectories of the particles. Since the current does not depend on time, the current lines remain unchanged. It also makes it easier to visualize the process. If the flow is unsteady, then the trajectories of the particles and the streamlines are different.

If the liquid is incompressible, then

$$\rho = const,$$

where $\rho$ — the density of the liquid.

If the movement is stationary (27), then

$$\frac{\partial V}{\partial t} = 0.$$

For this model problem, the solution area is a square with dimensions $x \in [0; 1], y \in [0; 1]$.

The structure of neural networks allows us to calculate analytically the gradient of the error functional. This problem was solved in two ways: with analytical calculation of the gradient of the error functional and with the use of automatic differentiation. Analytical calculation allowed to speed up the work of the program. This article presents the results of the solution obtained using the analytical calculation of the gradient of the error functional. Each variable had its own network of radial basis functions based on physics. All three networks are connected by a common root-mean-square

error functional. The functional consists of the sum of the squares of the residuals. To simplify the expression, the multiplier $\frac{1}{2}$ has been added for the gradient of the functional. The network was trained to the value of the total error functional equal to $10^{-4}$. The error functional can be written as

$$I = \frac{1}{2} \sum_{i=1}^{N} \left( u_i \frac{\partial u_i}{\partial x} + v_i \frac{\partial u_i}{\partial y} + \frac{\partial p_i}{\partial x} - \frac{1}{Re} \left( \frac{\partial^2 u_i}{\partial x^2} + \frac{\partial^2 u_i}{\partial y^2} \right) \right)^2 +$$

$$\frac{1}{2} \sum_{i=1}^{N} \left( u_i \frac{\partial v_i}{\partial x} + v_i \frac{\partial v_i}{\partial y} + \frac{\partial p_i}{\partial y} - \frac{1}{Re} \left( \frac{\partial^2 v_i}{\partial x^2} + \frac{\partial^2 v_i}{\partial y^2} \right) \right)^2 +$$

$$\frac{1}{2} \sum_{i=1}^{N} \left( \frac{\partial u_i}{\partial x} + \frac{\partial v_i}{\partial y} \right)^2 + \frac{\lambda}{2} \sum_{j=1}^{K} (u_j - u_j^A)^2$$

$$+ \frac{\lambda}{2} \sum_{j=1}^{K} (v_j - v_j^A)^2 + \frac{\lambda}{2} \sum_{j=1}^{K} (p_j - p_j^A)^2$$

where $Re$ – Reynolds number, $N$ – number of internal test points, $K$ – number of boundary test points, $\lambda$ – penalty multiplier, $u_j^A, v_j^A, p_j^A$ – analytical value of the $j$th test point at the boundary of the region.

The model is implemented in the MATLAB environment. The centers of the radial basis functions of each network were located on a uniform grid. The weight and width vectors were initialized with zero values. Experiments were conducted for each network to select the optimal number of neurons and trial points. The number of neurons for each physics-informed radial basis function networks is 64. The number of interior trial points for each network is 100. The number of boundary trial points for each network is 40. The coordinates of collocation points were generated as random numbers that were uniformly distributed in the solution domain. The network was trained using Nesterov's method. Only the weights were customizable. The location of the test points is shown in Figure 2.

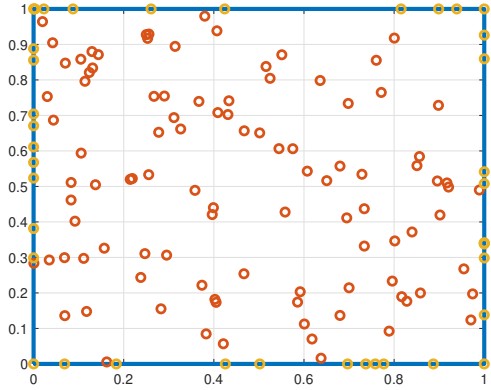

Figure 2: Test points

Let us analyze the effectiveness of the Nesterov method using the example of a model problem simulating the Kovasznay flow. The Nesterov method uses the history of parameter changes and the idea of the pulse accumulation method (28; 22). The formula for calculating the parameters in this case takes the form

$$\triangle \Theta^{(k+1)} = \alpha \triangle \Theta^{(k)} - \gamma grad I(\Theta^{(k)} - \alpha \triangle \Theta^{(k)}).$$

where $\Theta$ — the vector of one of the network parameters (we can consider the vector of all parameters), $\gamma$ — the selected numerical coefficient (learning rate), $\alpha$ — the moment coefficient, which takes values in the interval $[-0.5; 1.5]$.

Figure 3 shows the behavior of the error functional.

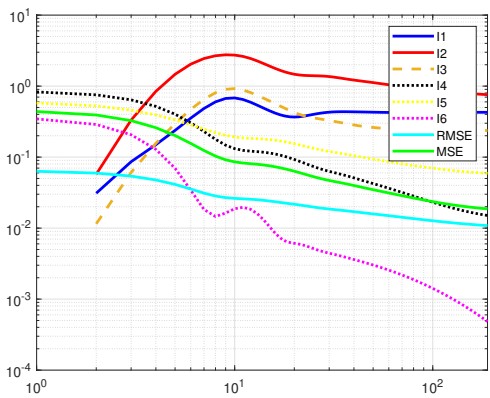

Figure 3: Error functional

$I1, I2, I3$ are the error values for the three Navier-Stokes equations, respectively. $I4, I5, I6$ are the error values for the three boundary conditions, respectively. The graph also shows the mean square error functional values (MSE) and the root mean square error functional values (RMSE). The root mean square error functional value of $10^{-2}$ is achieved in on average of 150 iterations. This problem was solved on a fully connected network (29). Physics-informed radial basis function networks are as accurate as fully connected networks, but they are easier and faster to train. Physics-informed radial basis function networks allowed us to achieve a root mean square error functional value of $10^{-2}$ on average in 15 minutes.

Plots of analytical solutions for velocity and pressure components are presented in Figures 4, 6, 8. The graphs obtained as a result of the network operation are shown in Figures 5, 7, 9. They have a visual similarity to the analytical solution. The value of the root mean square error functional for the first velocity component is on average $10^{-6}$. The value of the root mean square error functional for the second velocity component is on average $10^{-6}$. The value of the root mean square error functional for pressure averages $10^{-4}$. This is a testament to the effectiveness of the program.

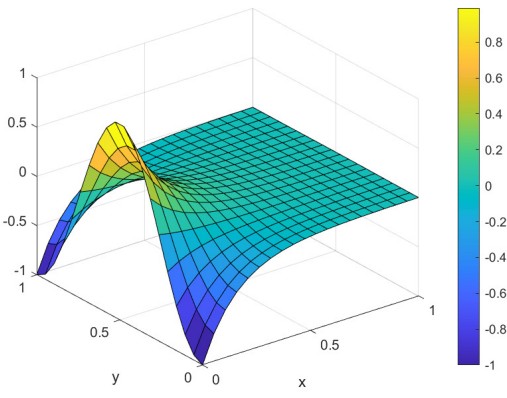

Figure 4: Analytical solution for the first velocity component

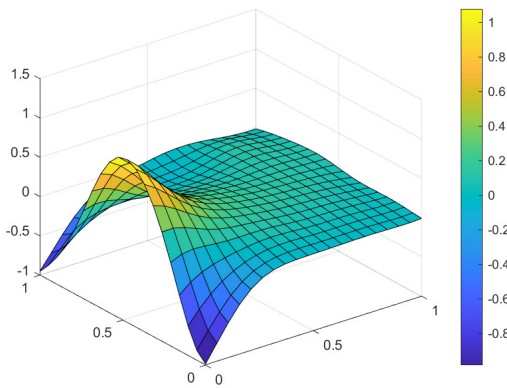

Figure 5: The first component of speed resulting from network operation

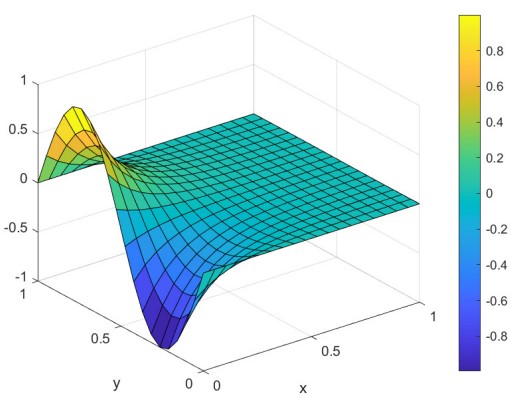

Figure 6: Analytical solution for the second velocity component

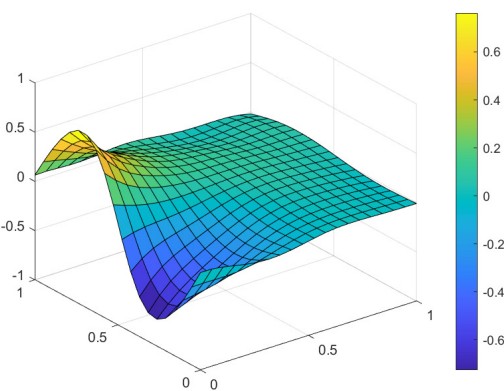

Figure 7: The second component of speed obtained as a result of network operation

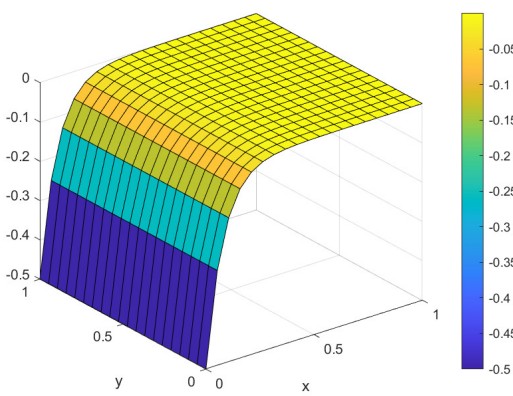

Figure 8: Analytical solution for pressure

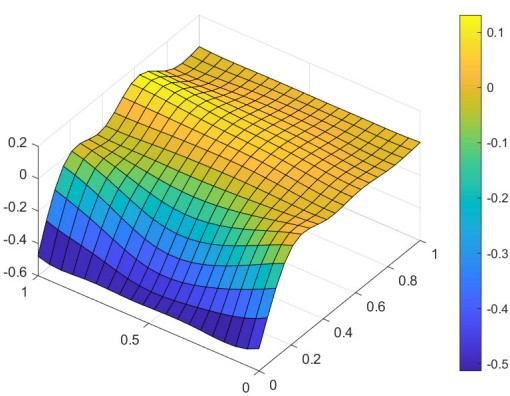

Figure 9: Pressure obtained as a result of network operation

Error graphs compared with the analytical solution for the two components of velocity and pressure are shown in Figures 10, 11, 12.

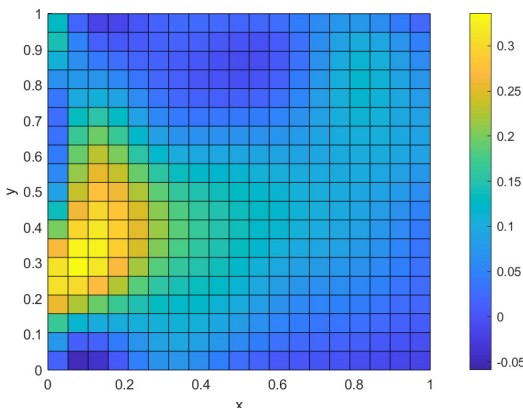

Figure 10: Error compared to the analytical solution for the first speed component

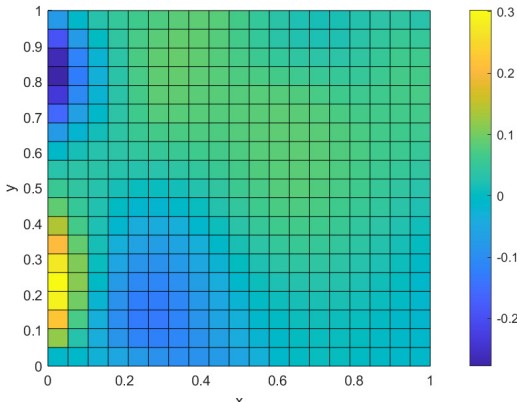

Figure 11: Error compared to the analytical solution for the second speed component

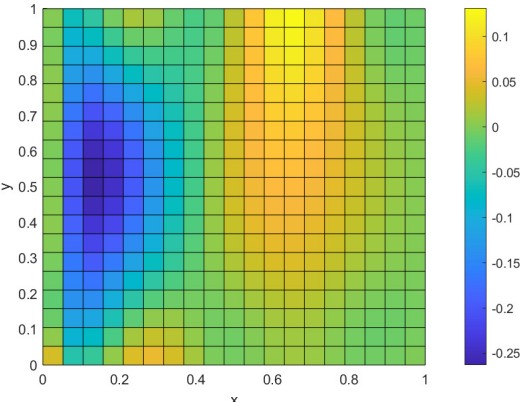

Figure 12: Error compared to the analytical solution for pressure

## 4 CONCLUSION

Algorithms and programs for solving direct and inverse boundary value problems on physically informed networks of radial basis functions have been developed. An algorithm and programs for solving a system of differential equations for solving two-dimensional stationary Navier-Stokes equations using networks of radial basis functions have been developed. The gradient descent method, Nesterov's method, Levenberg-Marquardt method, and Hessian diagonal approximation method have been adapted to solve the problems. Further development of the work involves developing an extension for one of the neural network implementation libraries in order to implement physically informed networks of radial basis functions using automatic differentiation. This will expand the possibilities of using physically informed networks of radial basis functions and will allow objectively comparing physically informed neural networks of different architectures.

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
