# OpenReview forum: "PHYSICS-INFORMED RADIAL BASIS FUNCTION NETWORKS AND KOLMOGOROV-ARNOLD NETWORKS"
_mathai.club/MathAI/2025/Conference — MathAI 2025 Oral_

### Official Review · Reviewer_XGkb · 2025-02-26
**Good paper, but without comparison**

**Rating:** 7
**Confidence:** 5

**Review:**

The paper proposes rather a fresh point-of-view on RBF as a KAN architecture. However, there is no comparison against a classical PINN architecture. I would like to mention that overall using RBF in PINNs is not novel. For example see [1,2].

[1] Xiao, Y., Yang, L. M., Du, Y. J., Song, Y. X., & Shu, C. (2023). Radial basis function-differential quadrature-based physics-informed neural network for steady incompressible flows. Physics of Fluids, 35(7).

[2] Bai, J., Liu, G. R., Gupta, A., Alzubaidi, L., Feng, X. Q., & Gu, Y. (2023). Physics-informed radial basis network (PIRBN): A local approximating neural network for solving nonlinear partial differential equations. Computer Methods in Applied Mechanics and Engineering, 415, 116290.

Maybe, we colud extract something when consider RBF in PINN as KAN-type network, but it is not clearly stated in the paper what the advantages are.

**Strengths:**
- Fresh point-of-view on RBF
- Strong mathematical formulation
- Experimental validation

**Weaknesses:**

- Lack of comparison against either analogues or DeepXDE
- Lack of consideration of KAN advantages
- Lack of benchmark experiments (PINNacle, for example)

Overall, I think that the paper could be presented at the conference and mentioned points are just for consideration for future work directions.

---

### Official Review · Reviewer_WiCP · 2025-02-27
**Review of Physics-informed radial basis function networks as Kolmogorov-Arnold networks**

**Rating:** 6
**Confidence:** 2

**Review:**

The authors propose a method to train physics-informed radial basis function networks for solving forward and inverse problems describing processes in piecewise homogeneous media. The paper stressed that their significant contribution is the tuning of the parameters of the basis functions during network training. However, this method is not novel, see references [14], [16] in the same paper. I suggest putting this paper in the context of the work done in [14] and [16].

[14] D.A. Stenkin V.I. Gorbachenko. Physics-informed radial basis function networks: Solving inverse problems for partial differential equations. Cyber-Physical Systems and Control II. CPSC 2021. Lecture Notes in Networks and Systems, 460:3–12, 2023.

[16] D.A. Stenkin V.I. Gorbachenko. Physics-informed radial basis-function networks. Technical Physics, 68:151–157, 2023.

---

### Official Review · Reviewer_Xc19 · 2025-02-27
**The paper is not well-written and contains misleading claims; I don't see any interesting or important results.**

**Rating:** 4
**Confidence:** 4

**Review:**

**Contribution of the paper**

The authors consider several PDE problems (some versions of the Poisson equation, two–dimensional stationary Navier-Stokes) and fit their solutions using a radial basis function model and a few standard optimization algorithms (gradient descent, Levenberg-Marquardt, Nesterov momentum). Along the way, the authors document some observations regarding the parameters of the model, number of iterations, etc. In the case of Navier-Stokes, the numerical solution is compared with the analytical one and this comparison is illustrated in a few figures.

**Strengths**

The paper provides some information about relative performance of different model fitting methods on the considered tasks.

**Weaknesses**

The paper reads like an unstructured informal log of a series of numerical experiments rather than a proper research paper. The results are not systematized in any way.

There are no theoretical contributions, and I don't see any conceptual novelty in this work.

I don't see any findings that would be of importance or significant interest to the wide audience.

The paper is misleading in claiming connection to Kolmogorov-Arnold networks. The model used by the authors (line 97) is a simple RBF models, i.e. a linear combination of shifted radial functions. It has nothing to do with the Kolmogorov-Arnold theorem, which requires two nonlinear layers implemented by univariate functions much more complex than Gaussians.

The clarity of writing is poor. Already at the very start, there is a mess with the notation. The RBF model is introduced in line 97 as a function of two arguments $x, y$ involving RBFs $\phi_k(x,y)$. Immediately below, in line 100, $\phi_k$ is written as a function of single argument, $\phi_k(x)$.  Further below, in line 107, the equation $Lu(\mathbf x) = f (\mathbf x), x ∈ Ω$ confusingly contains both roman $x$ and boldface $\mathbf x$.

---

### Official Review · Reviewer_Tzj8 · 2025-02-28
**A paper evaluating RBF networks on PIML problems. The novelty and the claimed KAN connection are questionable.**

**Rating:** 6
**Confidence:** 4

**Review:**

### Overview

This paper asserts the connection of RBF networks with KANs and evaluates the performance of RBF networks on practical tasks of direct (PDE solution) and inverse (PDE identification) boundary value problems. Compared to deep or fully connected networks, this approach leads to smaller model sizes which makes analytical differentiation and second-order optimization methods practical.

The authors claim that this work introduces the approach of adjusting not only the weights but also the parameters of the RBFs. But the previous works such as [14] already employ this approach. Therefore, the theoretical contribution of the paper is questionable.

Also, experiments don’t compare the quality (and the mentioned performance) of the method with alternatives, the only comparison made is with the ground-truth solution for the Kovazhny current case.

The discussion of the advantages or disadvantages of KAN-based (and this particular architecture) is lacking. The same is true about limitations and applicability (an obvious take is that networks with constant depth typically require an exponential layer width for approximating high-dimensional or structurally complex functions). Hence, the method is presented „in vacuum“: without the necessary context.

### Connection to KANs

The claimed connection to Kolmogorov-Arnold’s representation theorem isn't established. The small number of layers indeed matches but the fundamental idea of using univariate functions isn’t captured, since according to the formula at line 104, bivariate RBFs are used, whereas in the original KAN paper the transformations are univariate indeed and are represented as splines and followed by the linear combinations of their results as the only interaction between cooredinates. FastKAN [21] also uses RBFs in a one-dimensional context. So, the alleged relation to KANs requires clarification.

### Typesetting corrections

Generally, parentheses are often redundant or the wrong size ( `\left( … \right)` construct is recommended).

120: consider using the term „error funсtional“ instead of „error funсtionality“

174: according to the context, f_1 should be replaced by f_2

181-182: use a longer line to designate restriction

254: the reference is invalid

302: the reference is invalid

305: redundant parentheses

---

### Decision · Program_Chairs · 2025-03-08

**Decision:**

Accept (Oral)

**Comment:**

Your article has been accepted and you can make a presentation on the article. All articles will be sorted by rating and within the available conference places one author from each article will be invited. If there are not enough places, then you will either have the opportunity to present remotely or come at your own expense!